# Inorganic Salts and Antimicrobial Photodynamic Therapy: Mechanistic Conundrums?

**DOI:** 10.3390/molecules23123190

**Published:** 2018-12-03

**Authors:** Michael R. Hamblin, Heidi Abrahamse

**Affiliations:** 1Wellman Center for Photomedicine, Massachusetts General Hospital, Boston, MA 02114, USA; 2Department of Dermatology, Harvard Medical School, Boston, MA 02115, USA; 3Harvard-MIT Division of Health Sciences and Technology, Cambridge, MA 02139, USA; 4Laser Research Centre, Faculty of Health Science, University of Johannesburg, Johannesburg, Doornfontein 2028, South Africa; habrahamse@uj.ac.za

**Keywords:** antimicrobial photodynamic inactivation, potentiation by inorganic salts, sodium azide, potassium iodide, potassium bromide, potassium thiocyanate, potassium selenocyanate, sodium nitrite

## Abstract

We have recently discovered that the photodynamic action of many different photosensitizers (PSs) can be dramatically potentiated by addition of a solution containing a range of different inorganic salts. Most of these studies have centered around antimicrobial photodynamic inactivation that kills Gram-negative and Gram-positive bacteria in suspension. Addition of non-toxic water-soluble salts during illumination can kill up to six additional logs of bacterial cells (one million-fold improvement). The PSs investigated range from those that undergo mainly Type I photochemical mechanisms (electron transfer to produce superoxide, hydrogen peroxide, and hydroxyl radicals), such as phenothiazinium dyes, fullerenes, and titanium dioxide, to those that are mainly Type II (energy transfer to produce singlet oxygen), such as porphyrins, and Rose Bengal. At one extreme of the salts is sodium azide, that quenches singlet oxygen but can produce azide radicals (presumed to be highly reactive) via electron transfer from photoexcited phenothiazinium dyes. Potassium iodide is oxidized to molecular iodine by both Type I and Type II PSs, but may also form reactive iodine species. Potassium bromide is oxidized to hypobromite, but only by titanium dioxide photocatalysis (Type I). Potassium thiocyanate appears to require a mixture of Type I and Type II photochemistry to first produce sulfite, that can then form the sulfur trioxide radical anion. Potassium selenocyanate can react with either Type I or Type II (or indeed with other oxidizing agents) to produce the semi-stable selenocyanogen (SCN)_2_. Finally, sodium nitrite may react with either Type I or Type II PSs to produce peroxynitrate (again, semi-stable) that can kill bacteria and nitrate tyrosine. Many of these salts (except azide) are non-toxic, and may be clinically applicable.

## 1. Introduction

The relentless rise in antibiotic resistance amongst pathogenic bacteria and fungi [1], has motivated a search for newer antimicrobial techniques, that will not only kill drug-resistant bacteria but, themselves, will not induce resistance to emerge [2]. One of these newer antimicrobial techniques is known as antimicrobial photodynamic inactivation (aPDI) [3,4]. aPDI is derived from a cancer therapy known as photodynamic therapy (PDY) that was discovered as long ago as 1900 [5]. Photodynamic therapy (PDT) relies on a non-toxic dye called a photosensitizer (PS), that can be excited by harmless visible light to a long-lived triplet state [6]. The triplet state can undergo one of two different photochemical reactions [7]. Firstly, the triplet PS can undergo an electron transfer reaction between some surrounding electron donor or electron acceptor molecule, to form a radical anion or a radical cation. These radical species can further react with oxygen to form superoxide (O_2_^−^•), hydrogen peroxide (H_2_O_2_), and hydroxyl radicals (HO•), called the Type I photochemical pathway. Secondly, it can undergo energy transfer with ground state triplet oxygen to form excited state singlet oxygen (^1^O_2_) (see Figure 1). All these reactive oxygen species (ROS) can oxidize lipids, proteins, and nucleic acids, leading to cell damage and death. 

In order to use aPDI as an alternative approach to treat drug-resistant infections, it is necessary to ensure that the treatment is selective for pathogenic bacteria or fungal cells while, at the same time, sparing the normal host cells. This is generally accomplished by three methods [8]. Firstly, the PS molecules should be designed to selectively bind to, and be taken up by, the pathogenic microorganisms, and not by the host cells. This is generally done by ensuring the PS compounds possess a pronounced cationic charge, which not only ensures strong binding to the anionic residues on the microbial cells, but also ensures that the PS can penetrate through the outer membrane permeability barrier of Gram-negative bacteria. Secondly, the time between administration of the PS and shining the light (drug–light interval) should be short (few minutes) because the uptake by host cells is time-dependent, while the binding to bacteria is rapid [8]. Thirdly, the PS should be topically applied or injected into the infected area, and not given intravenously or orally, as would be done if PDT were being used to treat tumors.

Many different types of PS structure have been tested for their ability to kill bacteria and fungi in vitro, whether growing as planktonic suspensions or as biofilms [9]. It should be pointed out that the requirement for a pronounced negative charge, mentioned above, only strictly applies to Gram-negative bacteria, while anionic and neutral dyes can be highly active against Gram-positive bacteria. Figure 2 shows a selection of the PS chemical structures that will be discussed further in this review.

Despite the high interest being shown in antimicrobial PDT (aPDT) as a new approach to treat localized infections, it has not prevented a variety of laboratories from investigating ingenious ways to make it even better. Some of these approaches that have been reported can be summarized as combining aPDT with traditional antimicrobial compounds [10], improving the targeting capacity of PS by conjugating them to monoclonal antibodies that recognize bacteria [11] or to peptides that penetrate bacteria [12]; or combining PS with silver nanoparticles [13]. In the present review, we will discuss an entirely new approach to improving aPDI by combining the PS with simple non-toxic inorganic salts [14]. We will pay potential attention to attempting to explain the photochemical mechanisms and reactions involved in the oxidation of the inorganic ions. It should be pointed out that the concentration of inorganic salts that were used (10–400 mM) in these studies are not toxic to microbial cells after a 30 min incubation time. The original papers supplied this toxicity data.

## 2. Azide

We originally discovered the effect of inorganic salts on aPDI by working with sodium azide [15]. Sodium azide is frequently used as a classical physical quencher of singlet oxygen [16]. However, some workers have suggested that azide could also function as a chemical quencher of singlet oxygen-producing azide radicals [17]. The idea, in aPDI, is that the addition of azide to a mixture of bacteria and PS will inhibit the killing to a greater or lesser extent, depending on how much Type II photochemistry (singlet oxygen) or Type I photochemistry (radicals) there is with that particular PS. We showed that this hypothesis was indeed correct by comparing two different PS: a tris-cationic-buckminsterfullerene (BB6, Figure 2a), and a conjugate between polyethylenimine and chlorin(e6) (PEI-ce6, Figure 2b) [18]. PEI-ce6 mainly produced ^1^O_2_, since the killing was quenched by azide, while BB6 mainly produced Type I ROS, since the inhibition of bacterial killing was minimal.

We next looked at whether sodium azide (10 mM) would inhibit aPDI mediated by the phenothiazinium dye, methylene blue (MB, Figure 2c) [15]. To our surprise, the bacterial killing was not inhibited at all but, instead, was paradoxically potentiated (Figure 3). Further investigation identified the formation of azide radicals by spin trapping and electron spin resonance spectrometry. Interestingly, the azide radical was still formed in the absence of oxygen, and significant bacterial killing was still obtained in the absence of oxygen when azide was added, but not without azide (Figure 4). 

The hypothesized mechanism is a simple photo-induced electron transfer between MB and azide, as shown in Equation (1).
^3^PS + N_3_^−^ → PS^−^• + N_3_(1)

We went on to compare six different homologous phenothiazinium dyes: MB, toluidine blue O (TBO), new methylene blue (NMB), dimethylmethylene blue (DMMB), azure A (AA), and azure B (AB) [19]. We found both significant potentiation (up to 2 logs) and, also, significant inhibition (>3 logs) of killing by the addition of 10 mM azide, depending on Gram classification, washing the dye from the cells, and the precise dye structure (lipophilicity) as shown in Figure 5. Killing of *Escherichia coli* was potentiated with all 6 dyes after a wash, while *Staphylococcus aureus* killing was only potentiated by MB and TBO with a wash and DMMB with no wash. More lipophilic dyes with a higher log P value (octanol:water partition coefficient), such as DMMB, were more likely to show potentiation. We conclude that the Type I photochemical mechanism (potentiation with azide) likely depends on the microenvironment, i.e., higher binding of dye to bacteria. Bacterial dye-binding is thought to be higher with Gram-negative compared to Gram-positive bacteria, when unbound dye has been washed away, and with more lipophilic dyes.

We studied three decacationic fullerene monoadducts LC14, LC15 (Figure 2d), and LC16, excited by white or UVA light (340–380 nm) [20]. Bacterial killing was not much inhibited by the addition of azide anions and, in some combinations, was potentiated. In the absence of oxygen, microbial photokilling was highly potentiated (up to 5 logs) by the addition of azide anions (see Figure 6). We concluded that molecular functionality, that encourages Type I electron transfer, increases the ability of photoactivated fullerenes to kill microbial cells. Oxygen-independent photokilling is possible with fullerene monoadducts in the presence of azide anions, probably mediated by azidyl radicals. UVA excitation may kill bacteria partly by an electron transfer mechanism directly into bacteria, as well as by ROS. 

## 3. Iodide

Titanium dioxide (TiO_2_ known as titania) is a large band-gap semiconductor that is excited by UVA light [21]. When TiO_2_ nanoparticles with an average diameter of 25 nm (P25) [22] are mixed with bacteria, and excited by artificial UVA light or by natural sunlight, the bacteria are killed in a process known as antimicrobial photocatalysis [23]. These TiO_2_ NPs can produce ROS by two distinct mechanisms after photoexcitation. Absorption of a high-energy photon boosts electrons into the conductance band (Equation (2)). Firstly, these free electrons can transfer to oxygen to reduce it to superoxide radicals (Equation (3)). Secondly, the positive holes (h^+^) that are left in the TiO_2_ can oxidize water to produce hydroxyl radicals (Equation (4)).
TiO_2_ + *hv* → h^+^ + e^−^(aq)(2)
e^−^(aq) + O_2_ → O_2_^−^•(3)
h^+^ + H_2_O → HO• + H^+^(4)

We reported that the addition of potassium iodide solution (10 mM or 100 mM) to suspensions of P25 TiO_2_ that were illuminated with 360 nm light could drastically potentiate the microbial killing [24]. Up to 6 logs additional killing of Gram-positive bacteria, Gram-negative bacteria, and fungi was obtained (Figure 7). The microbial killing depended on the concentration of TiO_2_, the fluence of UVA light, and the concentration of potassium iodide (KI) (the best effect was at 100 mM). There was formation of long-lived antimicrobial species (probably hypoiodite and iodine) in the reaction mixture (detected by adding bacteria after light), but short-lived antibacterial reactive species (bacteria present during light) produced more killing. Tri-iodide (which has an absorption peak at 350 nm and forms a blue product with starch) was produced by TiO_2_–UVA–KI, but was much reduced when MRSA cells were also present. The model tyrosine substrate, *N*-acetyl tyrosine ethyl ester, was iodinated in a light dose-dependent manner.

We proposed that the mechanism involved oxidation of iodide anion by the holes produced during photocatalysis, in either a one-electron oxidation to form iodine radicals and then free iodine (Equation (5)), or in a two-electron oxidation to form iodonium cations and then hypoiodite (Equation (6)).
h^+^ + I^−^ → I•; 2I• → I_2_(5)
2h^+^ + I^−^ → I^+^ ; I^+^ + H_2_O → HOI + H^+^(6)

We then tested the phenothiazinium dye MB with KI [25]. Considering our finding with MB and azide [15], we initially supposed that Type I photochemistry would be required for the oxidation of iodide anion to bactericidal iodine species. Since we had used a concentration of 10 mM with sodium azide, we also used a concentration of 10 mM with potassium iodide [25] (see Figure 8). Although we achieved significant potentiation, the amount was not as high as was achieved later using higher KI concentrations. In this case, the mechanism appeared to be a combination of the production of stable antimicrobial species (free iodine and hypoiodite) and short-lived reactive species (iodine radicals). The concentration of KI turned out to be critical in determining the exact mechanism of microbial killing. If the concentration used is relatively low (up to 10 mM), then iodine radicals are mainly responsible but, if the KI concentration is increased up to 100 mM (or even higher up to 400 mM), then free iodine becomes mainly responsible. 

As mentioned above, we had previously believed that Type I photochemistry or, alternatively, photocatalysis, would be necessary to oxidize iodide to iodine. This assumption would change, however, when we investigated an anticancer PS called Photofrin (Figure 2e), which is a water-soluble porphyrin derivative usually injected intravenously, that has been shown to be completely unable to kill any Gram-negative bacteria when excited by light [26]. When 100 mM KI solution was added to suspensions of microbial cells (10^8^/mL) + PF (10 μM hematoporphyrin equivalent) + 415 nm light (10 J/cm^2^), we were able to eradicate (>6 log killing) five different Gram-negative species (*E. coli, Pseudomonas aeruginosa, Klebsiella pneumoniae, Proteus mirabilis,* and *Acinetobacter baumannii*), whereas no significant killing at all was obtained without KI [27] (see Figure 9). 

The mechanism of action appeared to be the generation of microbicidal molecular iodine (I_2_/I_3_^−^), as shown by comparable bacterial killing when cells were added to the mixture after completion of illumination, and light-dependent generation of free iodine, as detected by the formation of the starch complex. Gram-positive methicillin-resistant *S. aureus* is much more sensitive to aPDI (200–500 nM PF) and, in this case, potentiation by KI may be mediated mainly by short-lived iodine-reactive species. The fungal yeast *Candida albicans* displayed intermediate sensitivity to PF-mediated aPDI, and killing was also potentiated by KI. The reaction mechanism occurred via singlet oxygen (^1^O_2_). KI quenched ^1^O_2_ luminescence (1270 nm) at a rate constant of 9.2 × 10^5^ M^−1^ s^−1^. Oxygen consumption was increased when PF was illuminated in the presence of KI. Hydrogen peroxide (but not superoxide) was generated from illuminated PF in the presence of KI. Sodium azide completely inhibited the killing of *E. coli* with PF/blue light + KI. 

The Equations for the reaction of singlet oxygen with iodide are shown below (Equations (7)–(10)).
^1^O_2_ + 3I^−^ + 2H_2_O → I_3_^−^ + 2H_2_O_2_(7)

Considering possible reactive intermediates in these reactions, an interesting paper by Dalmazio et al. [28] from Brazil used mass spectrometry and ab initio free energy calculations to study the decomposition of hydrogen peroxide in the presence of iodide anions. They detected a species with *m*/*z* = 287 that was proposed to be HOOI_2_^−^ and explained its formation as depicted in Equations (5) and (10).
I^−^ + 2H_2_O_2_ + H^+^ → IOOH + 2H_2_O(8)
IOOH + I^−^ → HOOI_2_^−^(9)

The free energy calculations revealed that the most thermodynamically favored decomposition pathway of HOOI_2_^−^ was via Equation (11), to give two radicals, I_2_•^−^ and HOO•.
HOOI_2_^−^ → I_2_•^−^ + HOO•(10)

We next tested another PS that has been shown to be effective in killing Gram-positive bacteria, but not Gram-negative species, namely, the halogenated xanthene dye, Rose Bengal (RB, Figure 2f) [29]. Similar to the results described above with Photofrin, we found that addition of 100 mM KI potentiated green light (540 nm) mediated killing by up to 6 extra logs, of a broad spectrum of microbial species, including *E. coli, P. aeruginosa, S. aureus,* and *C. albicans*. Again, the mechanism was proposed to be singlet oxygen addition to iodide anion to form peroxyiodide, as described above. We also showed that addition of KI could potentiate RB-mediated photodynamic therapy in a mouse model of skin abrasions infected with bioluminescent *P. aeruginosa*. The use of bioluminescent bacteria and low-light imaging allows the non-invasive monitoring of the infectious burden in real time. As can be seen from Figure 10, only the combination of RB, KI, and light produced a dose-dependent loss of bioluminescence that was not seen with PDT alone (RB + light).

In an attempt to show the importance of PS–bacterial binding, we compared two differently charged porphyrins, 5,10,15,20-tetrakis(4-sulfonatophenyl)porphyrin dihydrochloride (TPPS4, Figure 2g, thought to be anionic and, therefore, not able to bind to Gram-negative bacteria) and tetra-(*N*-methyl-4-pyridyl)porphyrin (TMPyP4, Figure 2h, considered to be cationic and well able to bind to all bacteria) and produce some killing on its own, as well as eradication, when combined with KI (Figure 11A) [30]. As expected, TPPS4+light did not kill Gram-negative *E. coli* but, surprisingly when 100 mM KI was added, TPPS4 was highly effective (eradication at 10 µM + 10 J/cm^2^ of 415 nm light) (Figure 11B). TPPS4 was more effective than TMPyP4 in eradicating MRSA and *C. albicans* (regardless of KI). TPPS4 was also highly active against *E. coli* after a centrifugation step when KI was added, suggesting that the supposedly anionic porphyrin did, in fact, bind to bacteria (both Gram-positive and Gram-negative) and to *Candida*. This was confirmed by uptake experiments. We next compared the phthalocyanine tetrasulfonate derivative (ClAlPCS4, Figure 2i), which did not bind to bacteria or allow KI-mediated killing of *E. coli* after a spin, suggesting it was, indeed, truly anionic; see Figure 11C. We conclude that TPPS4 behaves as if it has some cationic character in the presence of bacteria, which may be related to its delivery from suppliers in the form of a dihydrochloride salt. 

## 4. Bromide

Because we had found that KI was so effective in potentiating aPDI, we wanted to test sodium bromide, a close analog of KI. However, when we tested several of the PSs that we had previously shown to undergo strong potentiation with KI addition (including PF, RB, MB, TPPS4, or TMPyP4), we were not able to detect any potentiation of killing of Gram-positive and Gram-negative bacteria, or fungi, by the addition of KBr. It was not until we tested the P25 TiO_2_ NPs (described in the previous section), for antimicrobial photocatalysis, that we found anything interesting [31]. When KBr (0–10 mM) was added to photoactivated TiO_2_ (P25), it potentiated the killing of Gram-positive (MRSA), Gram-negative (*E. coli*) bacteria and fungi (*C. albicans*), by up to 3 logs, as shown in Figure 12. We found that the mechanism of potentiation was probably due to generation of both short- and long-lived oxidized bromine species, including hypobromite, as shown by the following observations. There was some antimicrobial activity remaining in solution after switching off the light, that lasted for 30 min but not 2 h, and oxidized 3,3′,5,5′-tetramethylbenzidine. *N*-acetyl tyrosine ethyl ester was brominated in a light dose-dependent manner, however, no bromine or tribromide ion could be detected by spectrophotometry or LC-MS. The mechanism appears to have elements in common with the antimicrobial system (myeloperoxidase + hydrogen peroxide + bromide). Therefore, the mechanism can be clearly differentiated from the mechanisms applying to iodide, where it is quite clear that singlet oxygen is able to oxidize iodide to cytotoxic free iodine. Although hydroxyl radical has a redox potential (+2.31 V) that is more than enough to oxidize bromide (−0.78 V), singlet oxygen does not (+0.64 V) [32]. The minor inhibition of the TiO_2_-mediated photoactivation of singlet oxygen sensor green (SOSG) by bromide is probably caused by the physical quenching of ^1^O_2_, caused by its collision with bromide ions in a similar manner to that shown for iodide ions [33]. Therefore, we assumed that bromide oxidation was caused by a direct oxidation arising from the photoactivated TiO_2_, rather than oxidation by an intermediate oxygen-containing oxidizing species. 

When TiO_2_ is irradiated by UVA light, it acts as an oxidizing agent, due to the positive holes generated in the valence band by excitation of electrons into the valence band. Water can be oxidized to oxygen, showing that the oxidizing potential is at least +1.23 V. The redox potential of bromide to hypobromite is lower (−0.76 V) than the redox potential for bromide to bromine (−1.07 V), showing that bromide may be preferentially oxidized to hypobromite, rather than bromine.
2 H_2_O(l) → O_2_(g) + 4H^+^(aq) + 4e^−^ E^o^_ox_ = −1.23 V(11)
H_2_O + Br^−^ → 2e^−^ + 2H^+^ + BrO^−^ E^o^_ox_ = −0.76V(12)
2 Br^−^ → 2e^−^ + Br_2_ E^o^_ox_ = −1.07V(13)

However, there is another difference. The oxidation to hypobromite is a true 2-electron oxidation, while the oxidation to bromine is two separate 1-electron oxidations that proceed via the short-lived intermediate bromine radical:Br^−^ → Br• + e^−^.(14)

By contrast, the first step in the 2-electron oxidation to hypobromite is
Br^−^ → Br^+^ + 2e^−^.(15)

Bromide initially reacts to form the bromonium cation, that subsequently reacts with water to form hypobromite:Br^+^ + H_2_O → HOBr + H^+^.(16)

## 5. Thiocyanate

Pseudohalides are polyatomic analogues XY(Z)^−^ of halide anions that, in some cases, form dimeric molecules akin to halogens, called pseudohalogens. The chemistry of pseudohalogens resembles that of the true halogens, allowing them to substitute for halogens in several classes of chemical compounds [34]. We asked whether thiocyanate (SCN^−^) could potentiate the killing of *S. aureus* and *E. coli* [35]. Addition of KSCN (up to 10 mM) enhanced PDT killing (10 µM MB, 5 J/cm^2^ of 660 nm light) in a concentration-dependent manner. In the case of *S. aureus,* killing was increased by 2.5 log_10_ steps to a maximum of 4.2 log_10_ steps at 10 mM, and increased killing of *E. coli* by 3.6 log_10_ to a maximum of 5.0 log_10_ at 10 mM (see Figure 13). We determined that SCN^−^ rapidly depleted O_2_ from an irradiated MB system, reacting exclusively with ^1^O_2_, without quenching the MB excited triplet state. SCN^−^ reacted with ^1^O_2_, producing a sulfur trioxide radical anion (a sulfur-centered radical) that was demonstrated by EPR spin-trapping, by comparison with authentic spectra. We found that MB-PDT, with SCN^−^ present in solution, produced both sulfite and cyanide anions, and that addition of each of these salts separately enhanced MB-PDT killing of bacteria. We were unable to detect EPR signals of HO• which, together with kinetic data, strongly suggests that MB, known to produce HO• and ^1^O_2_, may, under the conditions used, preferentially form ^1^O_2_. 

The proposed mechanism of production of sulfur trioxide radical anion can be seen in Equations (17)–(19), involving both an addition reaction of ^1^O_2_ (Type II), as well a one-electron transfer to sulfite (Type I). This unusual sequence of reactions explains why purely Type II PSs (such as PF, RB, TPPS4) are not potentiated by KSCN. Apparently only PSs that can carry out a mixture of Type I and Type II mechanisms can be potentiated by KSCN. To confirm this hypothesis, we tested two other phenothiazinium dyes, namely, new methylene blue and dimethylmethylene blue [36]. Both of these dyes produced additional aPDI killing in the presence of KSCN, but RB and TPPS4 did not (data not shown) [36].
SCN + ^1^O_2_ → SO_2_CN^−^(17)
SO_2_CN^−^ + H_2_O → H_2_SO_3_ + CN^−^(18)
H_2_SO_3_ + e^−^ → SO_3_^−^• + 2H^+^(19)

## 6. Selenocyanate

Due to the success we obtained in potentiating aPDI with KSCN [35], we next tested another different pseudohalide salt, KSeCN [37]. We found that KSeCN (concentrations up to 100 mM) could also potentiate (up to 6 logs of killing) aPDI (*S. aureus* and *E. coli*) mediated by a number of different PSs: MB + red light, RB + green light, and TPPS4 + blue light. TPPS4 was effective at concentrations as low as 200 nM. When a mixture of selenocyanate with these PSs in solution was illuminated, and then bacteria were added after the light, there was up to 6 logs of killing (Gram-negative > Gram-positive) but the antibacterial species that was formed decayed rapidly (all gone by 20 min) (see Figure 14). Our hypothesis to explain this antibacterial activity was the formation of selenocyanogen (SeCN)_2_ by reaction of KSeCN with singlet oxygen (^1^O_2_), as shown by quenching of ^1^O_2_ by SeCN and increased photoconsumption of oxygen. The fact that lead tetraacetate reacted with SeCN (this is a literature preparation of (SeCN)_2_ [38,39]), to produce a short-lived antibacterial species, supported this hypothesis. 

The proposed reaction is as follows:2KSeCN + ^1^O_2_ + 2H^+^ → (SeCN)_2_ + H_2_O_2_.(20)

We were intrigued by the obvious difference in reaction mechanisms when thiocyanate and selenocyanate were compared as potentiating salts for aPDI. We, therefore, compared these two salts with light-activated PSs, and with other different oxidative reactions for killing Gram-positive and Gram-negative bacteria [36]. Overall, KSeCN was more powerful than KSCN, and worked with a wider range of PSs, while (as mentioned above) KSCN only worked with phenothiazinium salts. KSeCN produced killing when cells were added after light, suggesting production of a semi-stable species (SeCN)_2_, while KSCN did not. We tested three different oxidative reactions that each have the potential ability to kill bacteria: (a) lead tetraacetate; (b) Fenton reagent (hydrogen peroxide and ferrous sulfate); (c) hydrogen peroxide and horseradish peroxidase (H_2_O_2_/HRP). In every case, KSeCN was substantially more effective (several logs) than KSCN in potentiating the bacterial killing (see Figure 15). We concluded that (SeCN)_2_ is a mediator for aPDT using KSeCN, whereas sulfur trioxide radical anion is the mediator for aPDT using phenothiazinium salts potentiated by KSCN. For H_2_O_2_/HRP with KSCN, hypothiocyanite (OSCN^−^) has been proposed to function as the antibacterial agent in the literature [40,41,42], while hyposelenocyanite is said “not to exist” [43]. Lead tetraacetate is known to produce (SeCN)_2_ from KSeCN, as well as the analogous (SCN)_2_ from KSCN [44,45]. The mediators from Fenton reaction are unclear (pseudohalogen radical ions?). Both KSCN (which occurs naturally in the human body [46]) and KSeCN (that has been proposed as a dietary selenium supplement [47]) may be clinically applicable.

## 7. Nitrite

Although this data is, as yet, unpublished, we will briefly discuss potentiation of aPDI by the addition of sodium nitrite. It has been shown by Laura Pecci in Rome, Italy, that the photoactivation of some PS in the presence of nitrite can lead to nitration of tyrosine. One paper [48] used MB activated by red light, and another paper [49] used riboflavin activated by UVA light. In both cases, the mechanism of action was attributed to a Type 1 process that initially led to the simultaneous production of a tyrosine radical and a nitrogen dioxide radical, and that these two radicals then recombined to produce nitrotyrosine. 

We next tested whether the addition of sodium nitrite could potentiate aPDI killing of MRSA and *E. coli* bacteria (submitted for publication). Addition of up to 100 mM nitrite gave 6 logs of extra killing of *E. coli* in the case of Rose Bengal excited by green light, and 2 logs of extra killing in the case of MRSA using RB + green light. Comparable results were obtained for other PS (TPPS4 excited by blue light and MB excited by red light); see Figure 16. These findings suggest that nitrite reacts with PDT-produced ROS to produce a stronger oxidizing agent than singlet oxygen alone, and we hypothesized that peroxynitrate may be a candidate for this strong oxidizing agent. Some bacterial killing was obtained after light using a fullerene (LC15) + nitrite, and tyrosine ester amide was nitrated both “in” and “after” modes with all four PS. 

Peroxynitrate could be formed by a reaction between superoxide and nitrogen dioxide; formation of the latter was demonstrated by spin trapping with nitromethane. 

## 8. Conclusions

It has taken us considerable time to figure out the mechanistic complexities of the phenomenon of potentiation of aPDI by addition of various inorganic salts. Partly, this confusion was due to an initial assumption that Type I photochemistry was dominant in this effect. Since we had clearly established that Type I electron transfer was the mechanism responsible for potentiation with azide (necessarily so, because azide inhibits Type II energy transfer to singlet oxygen), we assumed this argument would also apply to other salts and, hence, we first tested TiO_2_ photocatalysis with iodide. This was successful, so we then tested iodide with MB, a dye well known to undergo Type I photochemistry, as shown by potentiation with azide [15]. This experiment again was successful, although perhaps the potentiation was not as strong as we first anticipated, using only 10 mM iodide [25]. We were quite surprised when we tested Photofrin with iodide, as we did not expect singlet oxygen to efficiently oxidize iodide [27]. However, by then, we had realized that we needed to use relatively high concentrations of KI (i.e., 100 mM instead of only 10 mM). The excellent potentiation observed with PF and iodide encouraged us to next test RB with iodide, and this combination was equally successful [29]. The experiments with the two porphyrins (TMPyP4 and TPPS4), although they uncovered interesting data on the actual charge borne on the tetrapyrroles, served to confirm the ability of ^1^O_2_ to oxidize iodide to produce antimicrobial species [30].

None of the other PSs we tested were able to oxidize bromide, therefore, only antimicrobial photocatalysis using TiO_2_ was able to be potentiated by bromide [31]. Moreover, the mechanisms of oxidation of halide anions appeared to be different between iodide and bromide, as well as between different PSs. In the case of photocatalysis, we detected formation of only hypobromite from bromide (no free bromine), and a mixture of free iodine and hypoiodite with iodide. By contrast, with other PSs (singlet oxygen-producing) there was no oxidation of bromide, and only free iodine was formed with iodide [24]. 

In the case of photoinduced electron transfer mechanisms with azide, it was possible to obtain oxygen-independent microbial killing, which could be explained by formation of cytotoxic azidyl radicals [15]. The fact that oxygen-independent bacterial killing was also observed with fullerenes (which are well known to undergo photoinduced electron transfer), confirmed this hypothesis [20]. 

Another interesting and initially confusing comparison was that between thiocyanate and selenocyanate. When it was found that only aPDI mediated by phenothiazinium dyes could be potentiated by thiocyanate, while PSs that are more likely to operate via a Type II mechanism were unaffected, this observation pointed us toward the dual role of Type II, followed by Type I, in producing a bactericidal species (sulfur trioxide radical anion) from thiocyanate. However, with selenocyanate, the story was different, and the action of singlet oxygen led to the formation of a semi-stable species that could only be provisionally identified as selenocyanogen, and which was supported by the formation of a similar acting semi-stable antimicrobial species formed from the reaction between KSeCN and Pb(OAc)_4_. However, KSCN did indeed appear to produce a semi-stable antibacterial species when reacted with Pb(OAc)_4_, although it was not as powerful as an antimicrobial agent as that formed from KSeCN and Pb(OAc)_4_. We confirmed that peroxidase/hydrogen peroxide was able to produce an antimicrobial species from KSCN, which has been widely attributed in the literature to be hypothiocyanite [40,41,42], but we also found that KSeCN could produce microbial killing from HRP/H_2_O_2_. It has been stated in the literature that “hyposelenocyanite does not exist”, although how it is possible to prove the non-existence of a compound is not entirely clear [43]. 

Yet another interesting mechanistic puzzle was provided by the case of nitrite. The data from Pecci et al. suggested that the tyrosine nitration reaction observed when nitrite was added to photoactivated MB [48] and RF [49], in the presence of tyrosine, depended on Type I photochemistry. However, the fact that killing was observed after light (at least in the case of photoactivated fullerenes) suggested that a semi-stable antimicrobial species was produced, in contrast to the radical species suggested by Pecci. 

Taken together, the teasing apart of the different reaction mechanisms, involved in the potentiation of aPDI by inorganic salts, has been a fascinating challenge. Not only has this research uncovered some novel reaction pathways, but it may also be of clinical relevance. Many of these salts occur naturally in the human body (nitrite and thiocyanate) or, else, possess low toxicity (iodide and selenocyanate). The fact that addition of iodide has been shown to be beneficial in multiple mouse models of localized infections is also highly encouraging.

## Figures and Tables

**Figure 1 molecules-23-03190-f001:**
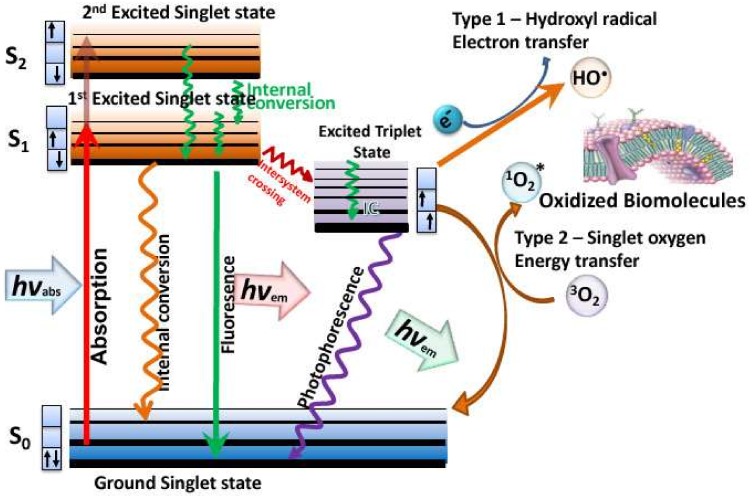
Jablonski diagram showing the photochemical mechanisms operating in antimicrobial photodynamic inactivation (aPDI).

**Figure 2 molecules-23-03190-f002:**
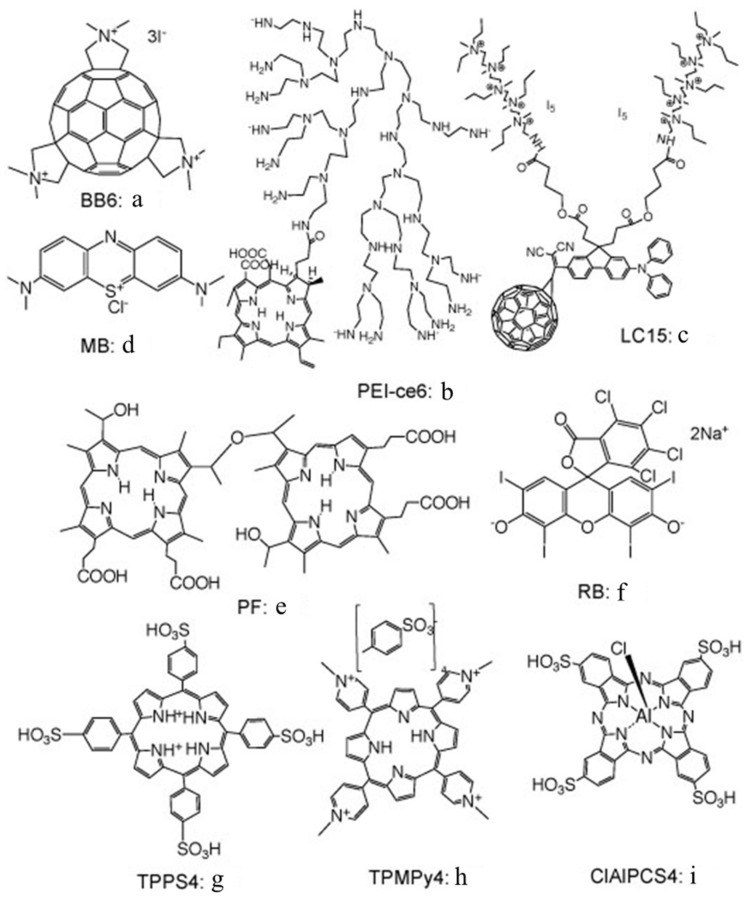
Chemical structures of several of the PS used in this paper. a, Triscationic fullerene BB6; b, conjugate between polyethylenimine and chlorin(e6) PEI-ce6; c, decacationic fullerene LC15; d, phenothiazinium methylene blue MB; e, hematoporphyrin derivative Photofrin PF; f, halogenated xanthene Rose Bengal RB; g, tetraanionic porphyrin TPPS4; h, tetracationic porphyrin TPMPy4; i, tetraanionic phthalocyanine ClAlPCS4.

**Figure 3 molecules-23-03190-f003:**
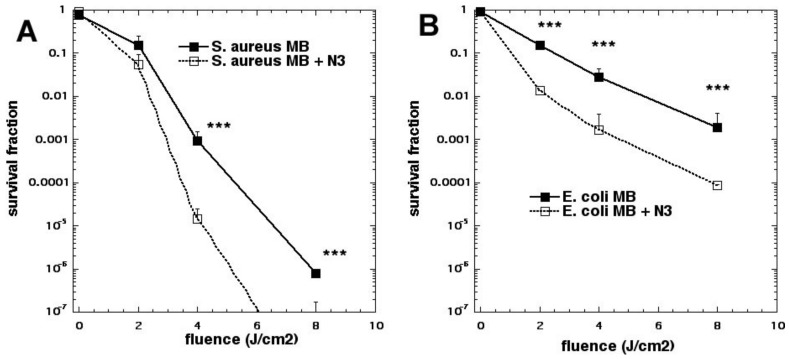
Paradoxical potentiation of methylene blue (MB) aPDI by addition of sodium azide. (**A**) *S. aureus*; (**B**) *E. coli.* Figure adapted from data contained in [15]. *** = *p* < 0.001.

**Figure 4 molecules-23-03190-f004:**
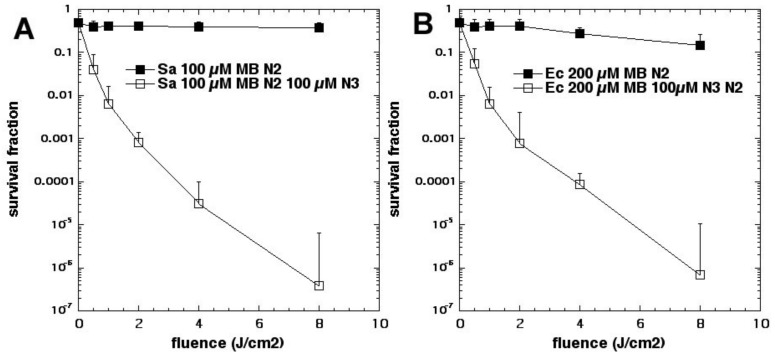
Oxygen-independent (under nitrogen) MB photoinactivation by addition of sodium azide. (**A**) *S. aureus*; (**B**) *E. coli.* Figure adapted from data contained in [15].

**Figure 5 molecules-23-03190-f005:**
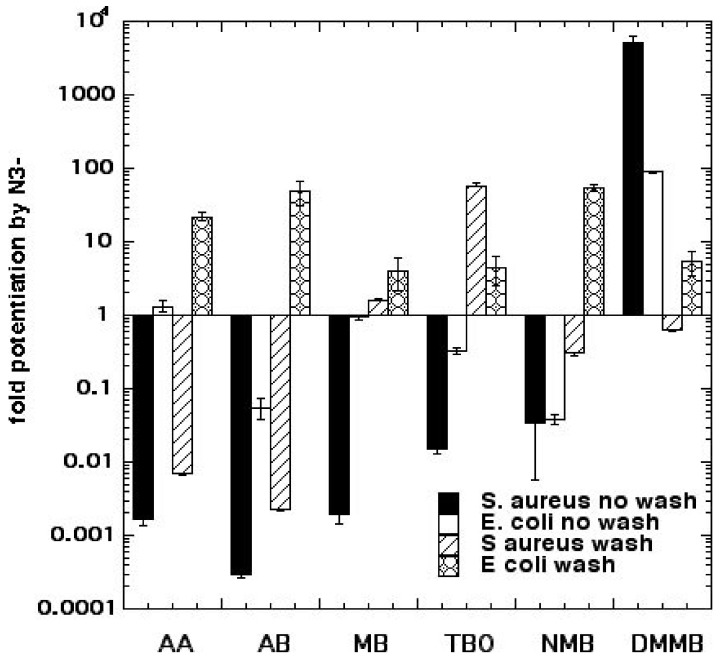
Comparison of six phenothizinium dyes for potentiation or inhibition of aPDI by sodium azide. Figure adapted from data contained in [19].

**Figure 6 molecules-23-03190-f006:**
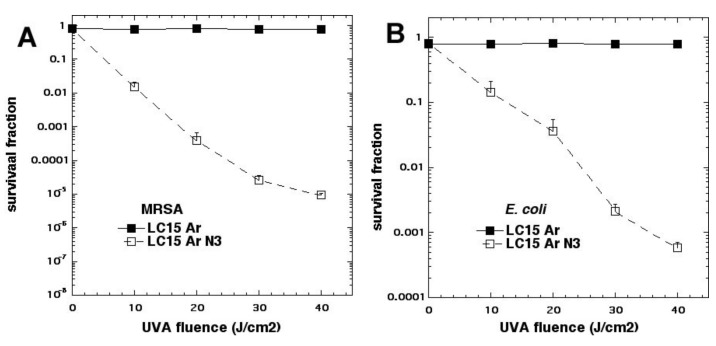
Oxygen-independent (under argon) LC15 fullerene photoinactivation by addition of sodium azide. (**A**) methicillin resistant *S. aureus* (MRSA); (**B**) *E. coli.* Figure adapted from data contained in [20].

**Figure 7 molecules-23-03190-f007:**
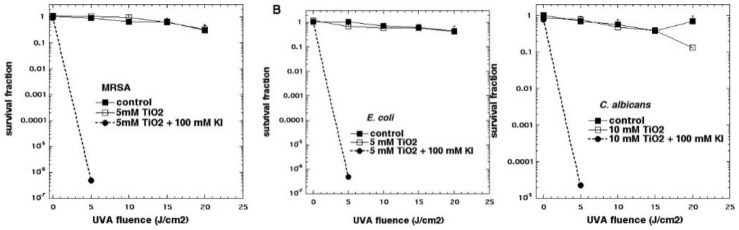
Titanium dioxide photocatalysis is potentiated by potassium iodide. (**A**) MRSA; (**B**) *E. coli*; (**C**) *C. albicans*. Figure adapted from data contained in [24].

**Figure 8 molecules-23-03190-f008:**
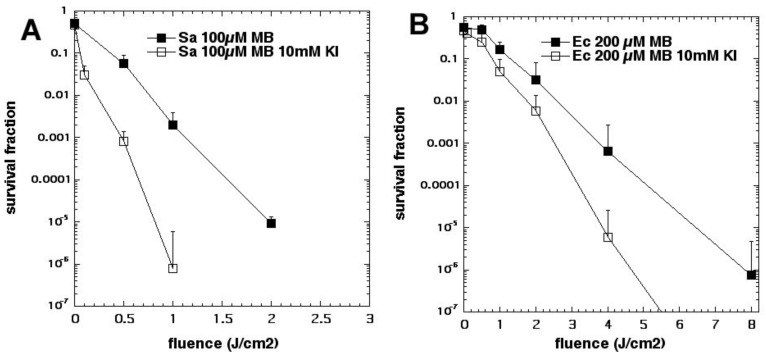
MB mediated aPDI is potentiated by potassium iodide. (**A**) *S. aureus*; (**B**) *E. coli.* Figure adapted from data contained in [25].

**Figure 9 molecules-23-03190-f009:**
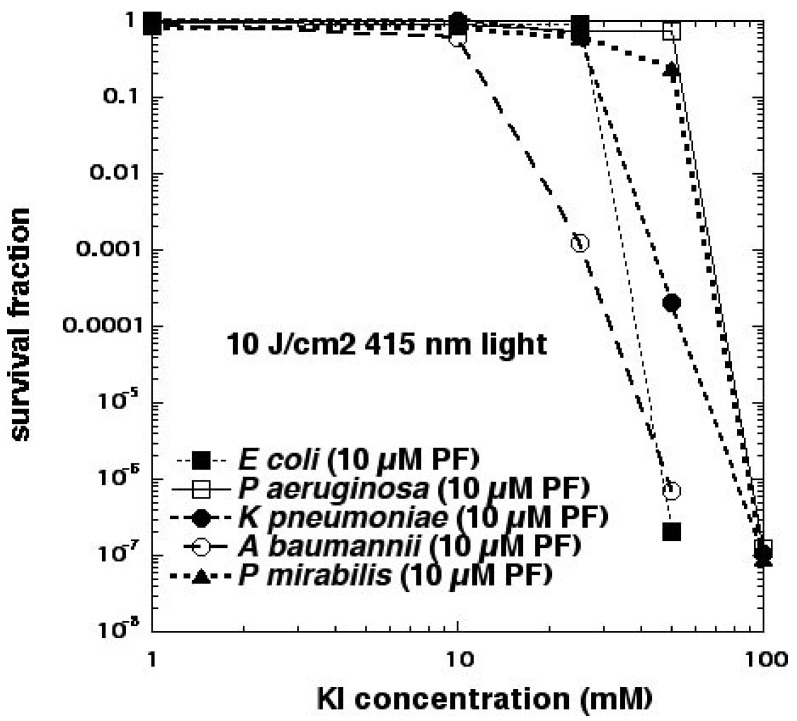
aPDI mediated by Photofrin kills Gram-negative bacteria in the presence of potassium iodide. Figure adapted from data contained in [27].

**Figure 10 molecules-23-03190-f010:**
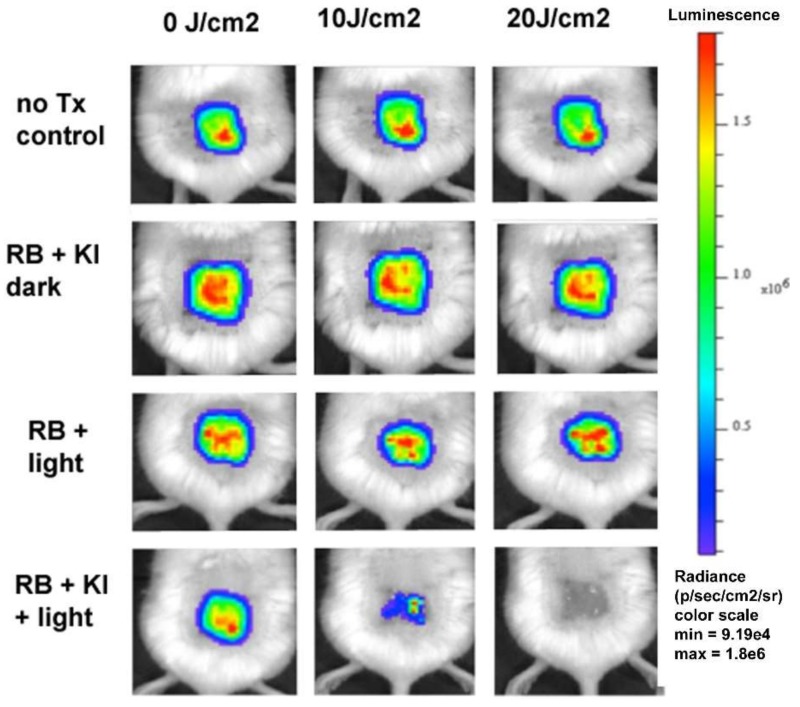
PDT using Rose Bengal plus potassium iodide can eradicate *P. aeruginosa* in a mouse infection model. Figure reproduced from [29], open access.

**Figure 11 molecules-23-03190-f011:**
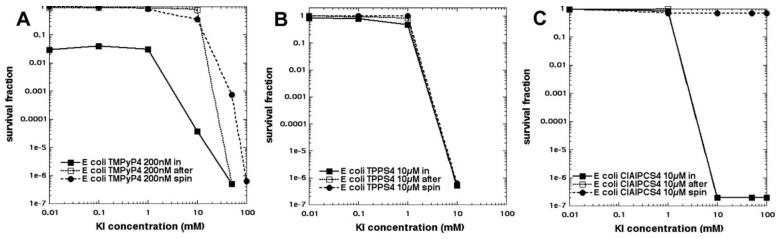
Potentiation of aPDI of *E. coli* with potassium iodide reveals that TPPS4 behaves like a cationic porphyrin. (**A**) TPMPy4; (**B**) TPPS4; (**C**) ClAlPCS4. Figure adapted from data contained in [30].

**Figure 12 molecules-23-03190-f012:**
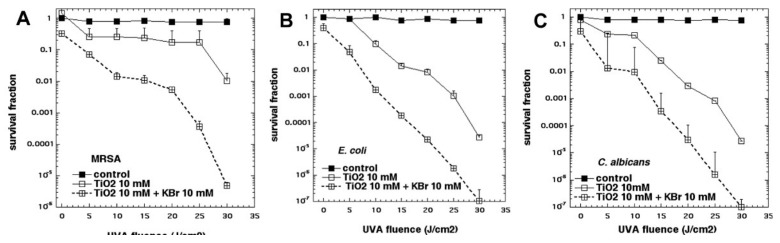
Titanium dioxide photocatalysis is potentiated by potassium bromide. (**A**) MRSA; (**B**) *E. coli*; (**C**) *C. albicans*. Figure adapted from data contained in [31].

**Figure 13 molecules-23-03190-f013:**
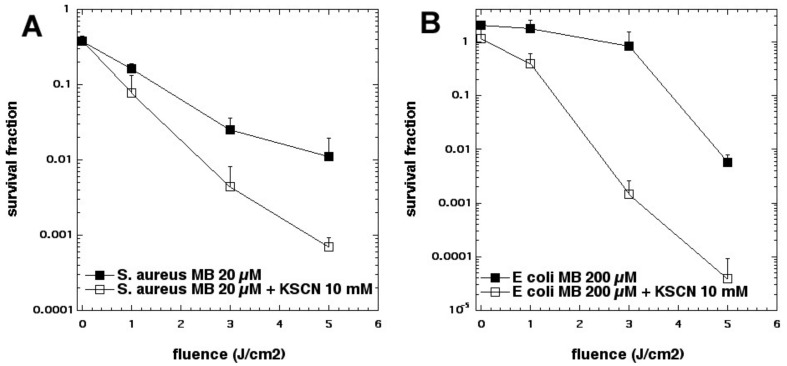
MB-mediated aPDI is potentiated by potassium thiocyanate. (**A**) *S. aureus*; (**B**) *E. coli*. Figure adapted from data contained in [35].

**Figure 14 molecules-23-03190-f014:**
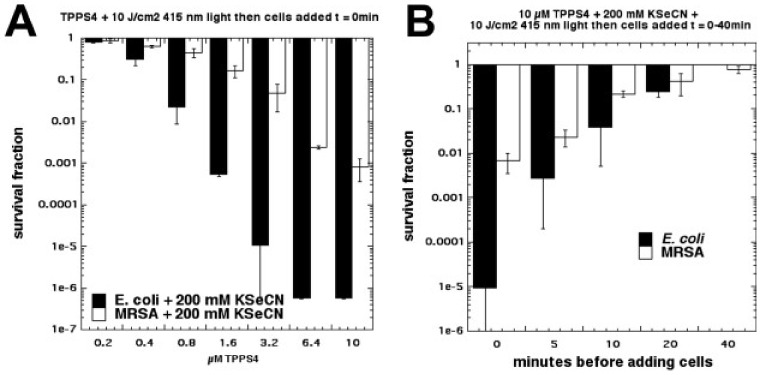
aPDI mediated by TPPS4 + potassium selenocyanate in “after” format. (**A**) Killing of MRSA and *E. coli* as a function of TPPS4 concentration; (**B**) Time-dependent decay of antibacterial activity of TPPS4 + KSeCN + light. Figure adapted from data contained in [37].

**Figure 15 molecules-23-03190-f015:**
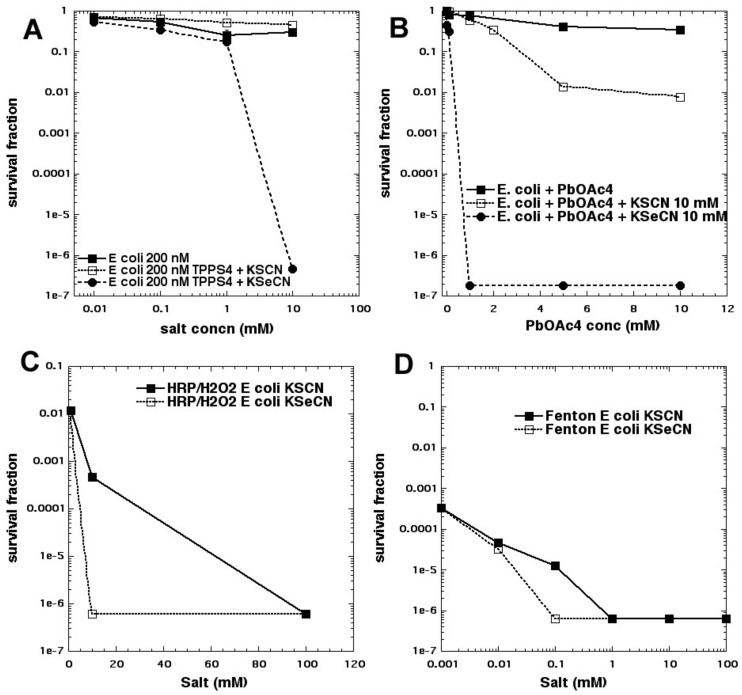
Comparison of thiocyanate and selenocyanate for potentiation of four different oxidative techniques against *E. coli*. (**A**) aPDI with TPPS4; (**B**) lead tetraacetate; (**C**) horseradish peroxidase and hydrogen peroxide; (**D**) Fenton reagent (Fe^2+^ + H_2_O_2_). Figure adapted from data contained in [36].

**Figure 16 molecules-23-03190-f016:**
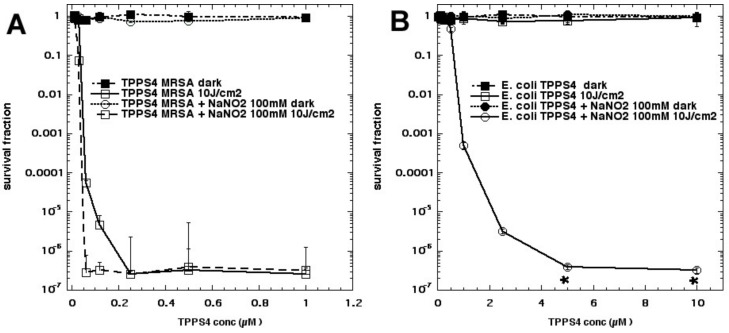
aPDI is potentiated by sodium nitrite. (**A**) MRSA; (**B**) *E. coli*. Unpublished data submitted for publication.

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
