# Peer review of "Inorganic Salts and Antimicrobial Photodynamic Therapy: Mechanistic Conundrums?"

_molecules, 2018, doi:10.3390/molecules23123190_

Reviewer 1 Report

The authors claim that the tested salts, except, azide, are non-toxic, but unfortunately no experimental proof is given – in all the figures showing the rise of photodynamic activity upon salt addition controls of the salt effect on bacteria are lacking. Such controls are really necessary especially since it is known that at least a part of the tested salts possesses antibacterial activity. For instance, sodium azide is known as a bacteriostatic preservative (see in: Russo I, Del Mese P, Viretto M, Doronzo G, Mattiello L, Trovati M, Anfossi G.

“Sodium azide, a bacteriostatic preservative contained in commercially available laboratory reagents, influences the responses of human platelets via the cGMP/PKG/VASP pathway.” Clin Biochem. 2008 41(4-5):343-9) and sodium nitrite possesses broad antimicrobial activity (see in: Anna C. Zemke, Mark T. Gladwin, Jennifer M. Bomberger, “Sodium Nitrite Blocks the Activity of Aminoglycosides against Pseudomonas aeruginosa Biofilms” Antimicrobial Agents and Chemotherapy 2015, 59, 3329-34), so the observed by the authors effect of bacterial killing could be due to salts only and not to photodynamic inactivation in the presence of salts.

Technical remarks.

1.      Line 28 – (SCN)2 – 2 in subscript

2.      Line 40 – PDT and not PDY

3.      Line 45 – “These reactive…” - a dot is missing before it

4.      Line 66 – “a pronounced negative charge mentioned above…” – please pay attention that on the line 57 you wrote about “a pronounced cationic charge”

5.      Line 90 – (singlet …) and not singley t

6.      Line 116 – log P – please define P

7.      Legends to Figs 6, 7, 8, 10, 11, 12, 13, 14, 15, 16 - to italicize bacteria names

8.      Line 217 – “logs. of a broad” – something is missing

9.      Line 281 TiO2 and Eq 11 –“2” and (l) – in a subscript

10.   Line 302 – 10mM – a space is missing

11.   Most of Figs - J/cm2 – “2” in superscript

12.   Line 365 – Fe+2 + H2O2 – fix super- and subscripts

13.   Fig 12 – in the A graph - there is no abscissa axis title  

14.    References are not formatted according to the Journal demands – why?

15.   Refs 1, 10, 29, 32, 36, 37, 43-45, 47 – publishing data are missing – either vol, year or pp, or all the data.

Author Response

The authors claim that the tested salts, except, azide, are non-toxic, but unfortunately no experimental proof is given – in all the figures showing the rise of photodynamic activity upon salt addition controls of the salt effect on bacteria are lacking. Such controls are really necessary especially since it is known that at least a part of the tested salts possesses antibacterial activity. For instance, sodium azide is known as a bacteriostatic preservative (see in: Russo I, Del Mese P, Viretto M, Doronzo G, Mattiello L, Trovati M, Anfossi G.

“Sodium azide, a bacteriostatic preservative contained in commercially available laboratory reagents, influences the responses of human platelets via the cGMP/PKG/VASP pathway.” Clin Biochem. 2008 41(4-5):343-9) and sodium nitrite possesses broad antimicrobial activity (see in: Anna C. Zemke, Mark T. Gladwin, Jennifer M. Bomberger, “Sodium Nitrite Blocks the Activity of Aminoglycosides against Pseudomonas aeruginosa Biofilms” Antimicrobial Agents and Chemotherapy 2015, 59, 3329-34), so the observed by the authors effect of bacterial killing could be due to salts only and not to photodynamic inactivation in the presence of salts.

Author response.

If the reviewer had looked up the original papers it would have been apparent that all the appropriate controls for microbial toxicity of the different inorganic salts had indeed been carried out. It appears that even what seem to be high concentrations of inorganic salts are not toxic to bacteria when the incubation time is only of the order of 30 minutes. This is very different from the situation where the bacteria are tested to see if they can grow in media containing high salts concentrations. The only exception is sodium azide, but here we limited the concentration to 10 mM. It is likely that 100 mM azide would have proved toxic even after 30 minutes.

Nevertheless we have added a sentence to the Introduction. “It should be pointed out that the concentration of inorganic salts that were used (10-400 mM) in these studies are not toxic to microbial cells after a 30 minute incubation time. The original papers supplied this toxicity data.”

Technical remarks.

General Revision. Figures 3-16 have been painstakingly redrawn to correct small errors in subscripts and superscripts and italics for bacterial names (but not for MRSA), which is an abbreviation.

1.     Line 28 – (SCN)2 – 2 in subscript

 Author response.

This has been corrected.

2.      Line 40 – PDT and not PDY

Author response.

This has been corrected.

3.      Line 45 – “These reactive…” - a dot is missing before it

Author response.

This is unclear

4.      Line 66 – “a pronounced negative charge mentioned above…” – please pay attention that on the line 57 you wrote about “a pronounced cationic charge”

Author response. This has been corrected to “pronounced positive charge mentioned above”

5.      Line 90 – (singlet …) and not singley t

Author response. This has been corrected

6.      Line 116 – log P – please define P

Author response. This has been changed to “More lipophilic dyes with a higher value of log P (octanol water partition coefficient),”

7.      Legends to Figs 6, 7, 8, 10, 11, 12, 13, 14, 15, 16 - to italicize bacteria names

Author response. This has been corrected

8.      Line 217 – “logs. of a broad” – something is missing

Author response. It has been changed to “we found that addition of 100 mM KI potentiated green light (540-nm)-mediated killing of a broad spectrum of microbial species, including E. coli, P. aeruginosa, S. aureus, and C. albicans, by up to 6 extra logs.”

9.      Line 281 TiO2 and Eq 11 –“2” and (l) – in a subscript

Author response. This has been corrected

10.   Line 302 – 10mM – a space is missing

Author response. This has been corrected

11.   Most of Figs - J/cm2 – “2” in superscript

Author response. This has been corrected

12.   Line 365 – Fe+2 + H2O2 – fix super- and subscripts

Author response. This has been corrected

13.   Fig 12 – in the A graph - there is no abscissa axis title

Author response. This has been corrected

14.    References are not formatted according to the Journal demands – why?

Author response. This will be done at proof stage

15.   Refs 1, 10, 29, 32, 36, 37, 43-45, 47 – publishing data are missing – either vol, year or pp, or all the data.

Author response. This will be done at proof stage

Reviewer 2 Report

The paper entitled: " inorganic salts and antimicrobial photodynamic therapy: mechanistic conundrums?", is a review exploring the mechanisms behind the modification of photodynamic antimicrobial action of different photosensitises in the presence of various inorganic salts in solution. This paper is interesting and well-written, adding interesting points to the present knowledge in the field.

Nevertheless, this reviewer find some minor issues that need to be addressed in the manuscript. First, to comply with instruction for authors, the inclusion of conflict of interest declaration is mandatory. Also, the manuscript writing needs improvement in the following locations:

L40 - Photodynamic therapy (PDT) and not PDY;

L45 - sentence needs full stop;

L61 - sentence needs an adequate citation;

L90 - needs spelling correction;

L99 - full stop is missing;

L211 - the equation number is incorrect;

L326 and 334 - needs punctuation correction;

L408 - needs spelling correction;

L421 - the ideia can be expressed with improved scientific language.

Author Response

The paper entitled: " inorganic salts and antimicrobial photodynamic therapy: mechanistic conundrums?", is a review exploring the mechanisms behind the modification of photodynamic antimicrobial action of different photosensitises in the presence of various inorganic salts in solution. This paper is interesting and well-written, adding interesting points to the present knowledge in the field.

Nevertheless, this reviewer find some minor issues that need to be addressed in the manuscript. First, to comply with instruction for authors, the inclusion of conflict of interest declaration is mandatory. Also, the manuscript writing needs improvement in the following locations:

L40 - Photodynamic therapy (PDT) and not PDY;

Author response. This has been corrected

L45 - sentence needs full stop;

Author response.

L61 - sentence needs an adequate citation;

Author response. We have added a citation.

L90 - needs spelling correction;

Author response. This has been corrected.

L99 - full stop is missing;

Author response. This has been corrected.

L211 - the equation number is incorrect;

Author response. This has been corrected.

L326 and 334 - needs punctuation correction;

Author response. This has been corrected

L408 - needs spelling correction;

Author response. This has been corrected.

L421 - the ideia can be expressed with improved scientific language.

Author response.

We have used the terms “both “in” and “after” modes” “ in the original paper so we will continue to use them.

Reviewer 3 Report

 The review titled “Inorganic salts and antimicrobial phytodynamic therapy: mechanistic condrums” is a nice and well written review of research using photosensitizers along with inorganic salts in the search for new antimicrobial compounds. It is my opinion that only minor changes are needed before acceptance.

One general note is that the review appears to be focusing on the authors work. Almost 40% of the references are from the authors.  This is also evident within the first paragraph of the conclusions where the authors use “we” 12 times.  While I am somewhat distant from this direct field, it makes me wonder if the authors are providing an accurate summary/representation of the literature or just their literature.  With that said, the authors are clearly leaders in their field.

Keywords: keyword should be different from those used in the title of the manuscript, also some of the current key words are more key phrases.

Ln 40: PDT (assuming photodynamic therapy) needs to be defined the first time it is used in the body of the manuscript

Ln 45: There does not appear to be a close to the parenthesis. I think the authors meant to include “)” after the hydroxy radical abbreviation?

Ln 45-47: The description of Type I photochemical pathway with the figure is not clear.  I would encourage the authors to provide a clear summary of their point. It is clear the authors know what they are trying to convey, but it is not necessarily clear to the reader.

Ln 59 and 61: A comma should be located after “secondly” and “thirdly”

Ln 65:  I would encourage authors to italicize “in vitro”

Ln 90: Did the authors mean a singlet oxygen instead of “ singley t oxygen”

Ln 127: Define UVA light

Author Response

The review titled “Inorganic salts and antimicrobial phytodynamic therapy: mechanistic condrums” is a nice and well written review of research using photosensitizers along with inorganic salts in the search for new antimicrobial compounds. It is my opinion that only minor changes are needed before acceptance.

One general note is that the review appears to be focusing on the authors work. Almost 40% of the references are from the authors.  This is also evident within the first paragraph of the conclusions where the authors use “we” 12 times.  While I am somewhat distant from this direct field, it makes me wonder if the authors are providing an accurate summary/representation of the literature or just their literature.  With that said, the authors are clearly leaders in their field.

Author response. I understand what the reviewers first impression is, but the stated goal of the article was to summarize our results in this relatively new area of potentiation of aPDI using inorganic salts. Since we first started exploring this phenomenon about six years ago, a few other groups have published papers confirming our results. Nevertheless I do not feel that any useful information would be added by including these few papers, particularly as we have carried out all the mechanistic investigations.

Keywords: keyword should be different from those used in the title of the manuscript, also some of the current key words are more key phrases.

Author response. We have removed duplication between the keywords and the title. It is common nowadays to use phrases in the keyword section.

Ln 40: PDT (assuming photodynamic therapy) needs to be defined the first time it is used in the body of the manuscript

Author response. This has been corrected

Ln 45: There does not appear to be a close to the parenthesis. I think the authors meant to include “)” after the hydroxy radical abbreviation?

Author response. The parentheses have been corrected.

Ln 45-47: The description of Type I photochemical pathway with the figure is not clear.  I would encourage the authors to provide a clear summary of their point. It is clear the authors know what they are trying to convey, but it is not necessarily clear to the reader.

Author response. We have rewritten the section to state “Firstly the triplet PS can undergo an electron transfer reaction between some surrounding electron donor or electron acceptor molecule, to form a radical anion or a radical cation. These radical species can further react with oxygen to form superoxide (O2-•), hydrogen peroxide (H2O2) and hydroxyl radicals (HO•), called the Type I photochemical pathway.

Ln 59 and 61: A comma should be located after “secondly” and “thirdly”

Author response. This has been corrected.

Ln 65:  I would encourage authors to italicize “in vitro”

Author response. This has been done

Ln 90: Did the authors mean a singlet oxygen instead of “ singley t oxygen”

Author response. This has been corrected.

Ln 127: Define UVA light

Author response. This has been done “UVA light (340-380 nm)”

Round  2

Reviewer 1 Report

The m/s can be published as is

Corrections in references are needed before the publication